# Quantum Molecular Resonance Inhibits NLRP3 Inflammasome/Nitrosative Stress and Promotes M1 to M2 Macrophage Polarization: Potential Therapeutic Effect in Osteoarthritis Model In Vitro

**DOI:** 10.3390/antiox12071358

**Published:** 2023-06-28

**Authors:** Teresa Paolucci, Vanessa Pino, Osama Elsallabi, Marialucia Gallorini, Gianantonio Pozzato, Alessandro Pozzato, Paola Lanuti, Victor Machado Reis, Mirko Pesce, Andrea Pantalone, Roberto Buda, Antonia Patruno

**Affiliations:** 1Department of Oral, Medical and Biotechnological Sciences, Physical Medicine and Rehabilitation, University G. D’Annunzio, 66100 Chieti, Italy; teresa.paolucci@unich.it; 2Department of Medicine and Aging Sciences, University “G. d’Annunzio” of Chieti-Pescara, 66100 Chieti, Italy; pivassa1@gmail.com (V.P.); osama.elsallabi@unich.it (O.E.); paola.lanuti@unich.it (P.L.); andrea.pantalone@unich.it (A.P.); roberto.buda@unich.it (R.B.); antonia.patruno@unich.it (A.P.); 3Institute on the Biology of Aging and Metabolism and Department of Biochemistry, Molecular Biology and Biophysics, University of Minnesota, Minneapolis, MN 55455, USA; 4Department of Pharmacy, University “G. d’Annunzio” of Chieti-Pescara, 66100 Chieti, Italy; marialucia.gallorini@unich.it; 5Telea Electronic Engineering srl, 36066 Sandrigo, Italy; gianantonio.pozzato@teleamedical.com (G.P.); alessandro.pozzato@teleamedical.com (A.P.); 6Research Centre in Sport Sciences, Health Sciences and Human Development, 5001-801 Vila Real, Portugal; vreis@utad.pt

**Keywords:** osteoarthritis, inflammation, macrophages, DAMP, QMR, rehabilitation

## Abstract

This study aimed to investigate the anti-inflammatory effects of Quantum Molecular Resonance (QMR) technology in an in vitro model of osteoarthritis-related inflammation. The study used THP-1-derived macrophages stimulated with lipopolysaccharide and hyaluronic acid fragments to induce the expression of inflammatory cytokines and nitrosative stress. QMR treatment inhibited COX-2 and iNOS protein expression and activity and reduced NF-κB activity. Furthermore, QMR treatment led to a significant reduction in peroxynitrite levels, reactive nitrogen species that can form during inflammatory conditions, and restored tyrosine nitration values to those similar to sham-exposed control cells. We also investigated the effect of QMR treatment on inflammasome activation and macrophage polarization in THP-1-derived macrophages. Results showed that QMR treatment significantly decreased NLRP3 and activated caspase-1 protein expression levels and downregulated IL-18 and IL-1β protein expression and secretion. Finally, our findings indicate that QMR treatment induces a switch in macrophage polarization from the M1 phenotype to the M2 phenotype.

## 1. Introduction

In the pathogenesis and progression of osteoarthritis (OA), a complex multi-tissue pathology, low-grade and chronic inflammation play a key role as evidenced by sensitive imaging methods (magnetic resonance imaging (MRI) and ultrasound) [1,2].

It has been established that in addition to the production of proinflammatory and destructive mediators, the cellular component and in particular the macrophages in the OA synovium are responsible for maintaining synovial inflammation. Studies have demonstrated that OA synovial macrophages can make up around 30–40% of the cellular content exhibiting an activated phenotype and producing proinflammatory cytokines like TNFα and IL-l and matrix metalloproteinases and expressing aggrecanases involved in driving the inflammatory response and joint destruction [3].

In the immune system, macrophages are antigen-presenting cells (APCs) with effective phagocytic activity responsive to cytokines released by lymphocytes. We know that macrophages evoke responses that can be variable and dependent on the tissue environment that influences their polarization. The pro-inflammatory phenotype, classically activated M1 macrophage, produces high levels of TNFα, IL-1, IL-6, IL-12, IL-23, reactive oxygen species (ROS), and low levels of IL-10. The anti-inflammatory phenotype or alternatively activated macrophage (M2), of which there are three subsets [4], produces high levels of IL-10, IL-1 receptor antagonist, decoy IL-1RII, TGFβ, and low levels of IL-12. Both phenotypes are necessary for the correct resolution of inflammation. The inflammatory signals in some pathological situations, such as AO, are activated both in the presence and in the absence of infection. Then, the mechanism by which macrophages monitor an inflamed environment is by sensing pathogen-associated molecular patterns (PAMPs) and endogenous danger-associated molecular patterns (DAMPs) leading to the assembly and activation of the inflammasome. Emerging evidence suggests that DAMPs-induced inflammation plays an important role in the pathogenesis of OA [5]. DAMPs or endogenous pattern recognition receptors (PRRs)-ligands are a group of molecules derived from host tissues or cells, either components of cells or induced gene products in specific conditions. The majority are extracellular matrix components such as fibronectin, heparan sulfate, biglycan, fibrinogen, oligosaccharides of hyaluronan, and hyaluronan breakdown fragments [6].

Extensive in vitro and in vivo data reported in the literature show the use of biophysical stimulation techniques to treat various diseases in human beings. Biophysical stimulation refers to the use of physical stimuli, such as electromagnetic fields, ultrasound, and mechanical vibrations, to promote healing and tissue regeneration. For example, electromagnetic field therapy has been used to treat conditions such as bone fractures, chronic pain, and OA [7]. Ultrasound has been used to treat soft tissue injuries, promote bone healing, and improve blood flow [8]. Mechanical vibrations, such as those used in whole-body vibration therapy, have been studied for their potential to improve muscle strength, bone density, and balance in elderly individuals [9]. Specifically, in rehabilitation, instrumental physical therapies can be classified according to the energy source: (i) electromagnetic (such as electric current and electromagnetic fields); (ii) mechanical (mechanical sound waves and vibratory energy); (iii) thermal (micro-waves and short waves). Each different instrumental physical therapy induces peculiar effects on biological tissues, some being more suitable for pain control, others for inflammation and for regeneration. Also, the most used methods for administrating physical energy to a biological system can be divided into (a) electrical energy applied directly to the tissue using adhesive electrodes (capacitively coupled electrical field), (b) electromagnetic energy applied using coils (pulsed electromagnetic fields, PEMFs), and (c) ultrasound energy applied directly to the tissue in the form of mechanical forces (low-intensity pulsed ultrasound system). The understanding of the interaction between physical agents and biological systems is particularly complex and depends on waveform, frequency, duration, and energy, on the identification of the dose–response effects, and on the characteristics of the targeted cell/tissue types. Obviously, the identification of the effects of physical agents in terms of how these can modulate a particular cellular function constitutes the basis for its possible clinical application. Electromagnetic fields are outlined as a potential alternative or in association with pharmacological treatments in several inflammatory-related pathologies [10,11,12,13,14,15]. Several papers have demonstrated the anti-inflammatory effect of PEMFs exposure in human synoviocytes, chondrocytes, and osteoblasts with a significant reduction in some of the most relevant pro-inflammatory cytokines [16,17].

In this study, the in vitro anti-inflammatory activity of Quantum Molecular Resonance (QMR) technology was investigated. Nowadays, applied mainly in bipolar coagulators and electrosurgery devices [18,19,20], QMR is a non-ionizing, low-potency technology that uses high-frequency waves in the range between 4 and 64 MHz delivered through alternating electric fields. QMR develops quanta of energy capable of breaking molecular bonds without increasing the kinetic energy of the affected molecules. This prevents the temperature from rising and limits damage to the surrounding tissue. Unlike medical devices with ELF technology, QMR produces nanosecond pulses that could penetrate through the plasma membrane and interact with cytoplasmic organelles [21]. To evaluate the potential mechanisms of action of QMR-therapeutic intervention for OA, in this study, we have used a physiologically relevant in vitro model of OA-related inflammation based on induction inflammatory cytokine expression in response to the combination of a PAMPs (lipopolysaccharide (LPS)) and a DAMPs (hyaluronan (HA) fragments) exposure.

In a previous study, it was demonstrated that small HA fragments (<289 kDa) can induce a macrophage-related inflammatory response in THP-1 cells when the cells are primed with the TLR4 agonist LPS, in a physiologically relevant concentration. This is important because the presence of LPS in the synovial fluid of OA patients has been associated with the presence of activated macrophages within the synovium and radiographic OA severity [22].

We examined the QMR effect, assessing the impact on inducible nitric oxide synthase (iNOS) and cyclooxygenase-2 (COX-2) expression/activity and peroxynitrite generation on differentiated (macrophages) human immortalized monocyte-like cell line (THP-1), a well-established cell model for the immune-modulation study. Considering the fact that the key response in vitro of macrophages to the stimulus by PAMPs and/or DAMPs, respectively, is the activation of the inflammasome formed by a pattern recognition receptor called NOD-like receptor pyrin domain containing three (NLRP3), we sought to determine the QMR anti-inflammatory responses through the NLRP3 signaling pathway.

Finally, we also tested whether QMR exposure could promote changes in the macrophage phenotypes related to M1/M2 macrophage polarization.

## 2. Materials and Methods

### 2.1. QMR Exposure System

Exposure was performed using a QMR generator supplied by Telea (Telea Electronic Engineering, Sandrigo, VI, Italy). The device generates alternating electric currents characterized by high-frequency (4–64 MHz) and low-intensity waves [23]. The QMR generator setup was with the following parameters: power supply, 230 V, ~50/60 Hz; maximum power input, 250 VA; and power output, 5 W/400 Ω. QMR was supplied using a pair of custom-made electrodes placed directly on the edge of a 100 mm Petri dish and connected to the QMR generator (Figure 1). No significant temperature changes were observed to be associated with the application of the QMR generator (∆T = 0.1 °C).

### 2.2. Cell Culture and Stimulation Protocol

The THP-1 cell line (ATCC, Rockville, MD, USA) was grown in RPMI (Roswell Park Memorial Institute) 1640 (Merck, Milan, Italy) supplemented with fetal bovine serum (10%) (Merck, Milan, Italy), L-glutamine (2 mM), and 10 mM HEPES (4-(2-hydroxy-ethyl)-1-piperazineethanesulfonic acid) (10 mM) in a humidified 5% CO_2_ incubator at 37 °C. The induction to THP-1 cell differentiation into mature monocyte-derived macrophages was carried out with 20 ng/mL of phorbol myristate acetate (PMA) (Merck, Milan, Italy) for 72 h [24,25].

For TLR pathways activation, differentiated (macrophage) THP-1 cells were incubated with a physiologically relevant concentration of LPS (10 ng/mL) from E. coli, Serotype R515 (Enzo Life Sciences), and ultra-low molecular weight (ULMW, 7.5 KDa) HA (10 mg/mL) (R&D Systems). After treatment with LPS/HA for 24 h, the planned experiments considered two modalities of temporal: (a) 24 h; (b) 4 days. The cells were then subjected to QMR (Quantum Magnetic Resonance) stimulation for 10 min per day, for 4 consecutive days, with a rest period of 24 h between treatments. One QMR setting was used, corresponding to nominal powers of 30.

Overall, the goal of the experiment was to recreate therapeutic conditions in vitro, in order to investigate the effects of QMR stimulation on macrophage cells. By testing different time settings and comparing them to sham-exposed controls, we assessed the potential benefits of QMR stimulation on cell cultures.

### 2.3. MTS Assay for Viability and Metabolic Cellular

The MTS assay is a colorimetric assay that measures the metabolic activity of cells. The assay is based on the ability of metabolically active cells to convert MTS into a colored formazan product, which is measured at 490 nm using a spectrophotometer.

MTS assay was used to measure the metabolic activity of THP-1-differentiated cells in response to an activating stimulus (LPS/HA) and treatment with QMR at two-time points (24 h and 4 days).

To perform the assay, cells were plated in 96-well plates, incubated with the activating stimulus alone, and exposed to QMR for the desired time period. After incubation, MTS was added to each well, and the cells were incubated for an additional 4 h. The absorbance of the formazan product was measured at 490 nm using a spectrophotometer, and the percentage of metabolically active cells was calculated based on the absorbance of treated samples relative to control cells. All assays were performed in triplicate to ensure the reproducibility of results.

### 2.4. Western Blot

Western blot analysis was performed to detect the protein expression levels of iNOS, nitrotyrosine, COX-2, p-NF-kB (p65), NF-kB (p65), IL-18, IL-1β, NLRP3, caspase-1 in macrophage cells. The cells were incubated at two-time points (24 h and 4 days) and were lysed in RIPA buffer. The protein concentration was determined by a bicinchoninic acid assay [26]. Briefly, total protein extracts were separated by sodium dodecyl sulfate-polyacrylamide gel electrophoresis (SDS-PAGE) on a 10% gel and then transferred to nitrocellulose membranes. The blots were probed with primary antibodies against the target proteins and incubated overnight. The primary antibodies used were anti-iNOS (NOS2 (N-20), sc-651), anti-COX-2 (ab52237), p-NF-kB (p65) (sc-166748), NF-kB (p65) (sc-8008), IL-18 (10663-1-AP), IL-1β (sc-32294), NLRP3 (D2P5E) (#13158), caspase-1 (AB_2016691). After washing, the blots were incubated with secondary antibodies conjugated with horseradish peroxidase. The protein expression levels were detected using Super Signal Ultra chemiluminescence detection reagents. β-actin was used as a loading control, and the densitometry analysis of the blots was performed using a gel analysis software package (Gel Doc 1000; Bio-Rad, Milan, Italy). The results were presented as mean values ± standard deviations (S.D.) of normalized densitometric values on the loading control.

### 2.5. Analysis of Peroxynitrite Generation

The cells were grown and treated with LPS/HA with or without exposure to QMR at two-time points (24 h and 4 days), in 6-well plates. The production of peroxynitrite was detected using a DAX-J2™ PON Green probe (Cell 205 MeterTM Fluorimetric Intracellular Peroxynitrite Assay Kit, AAT Bioquest, Pleasanton, CA, USA) and flow cytometry at 24 and 48 h. To perform the analysis, the exposure medium was removed, and a fresh medium containing 1 µL/mL of DAX-J2™ PON Green 500 X was added to the cells. The cells were then incubated at 37 °C and 5% CO_2_ in the dark for 1 h before processing according to the manufacturer’s instructions. The analysis was performed using a CytoFLEX Flow Cytometer equipped with a 488 nm laser and an FL1 (FITC) detector in a linear mode. The cells were gated based on their forward and side scatter properties (FSC/SSC), and the relative fluorescence emissions were analyzed using the CytExpert software (version 2.5). The results were expressed as mean fluorescence intensity (MFI) ratios on the unstained control.

### 2.6. NOS Activity

Oxyhemoglobin assay was used to monitor the intracellular NOS enzyme activity [27]. The assay was initiated with enzyme in a total volume of 1 mL. NO reacts with oxyhemoglobin to yield methemoglobin that is detected at 576 nm (e = 12,000 M^−1^ cm^−1^) using a Perkin-Elmer LamdaBIO UV–Vis spectrophotometer.

### 2.7. Prostaglandin PGE_2_ Measurement

The concentration of PGE_2_ was measured using a colorimetric detection method, specifically an enzyme-linked immunosorbent assay (ELISA) according to the instructions provided by the manufacturer of the assay kit (ADI-901-001). THP-1 cells were seeded onto six-well tissue culture plates at concentrations of 5 × 10^4^ mL^−1^ and cultured in the presence or absence of LPS/HA with or without QMR treatment. Aliquots of cell culture supernatants, human anti-PGE_2_, and PGE_2_ conjugate were added to the 96-well plates coated with anti-human antibody and incubated for 1 h at room temperature. Then, the reaction mixture was aspirated and 3,3′,5,5′-tetramethylbenzidine (TMB) enzyme substrate (150 μL) was added and incubated for 30 min at room temperature. The reaction was stopped by adding sulphuric acid (1 M), and the absorbance was measured at 405 nm.

### 2.8. Immunophenotyping In Vitro

The expression of surface markers (CDs) in macrophages was analyzed using flow cytometry. After differentiation, macrophages were stimulated with LPS and exposed to treatments for 4 days. After that, cells were harvested with Stem Pro™ Accutase™ cell dissociation reagent (ThermoFisher Scientific, Waltham, MA, USA), collected by centrifugation in the cold, and washed once with FACS buffer prepared with 10 mM 4-(2-hydroxyethyl)-1-piperazineethanesulfonic acid (HEPES) buffer at pH 7.4, 140 mM sodium chloride (NaCl), and 2.5 mM calcium chloride (CaCl_2_). Cells were incubated with fluorochrome-conjugated antibodies (1:50 dilutions) in 50 μL of FACS buffer for 15 min in the dark. Cells were stained separately in each single screening tube with a cluster of differentiation (CD)80-PE and CD163-PE (all purchased by BD Biosciences, MA, USA) Then, the excess antibodies were removed by adding fresh FACS buffer and centrifugation. After that, 20,000 events were run in a Beckman Coulter CytoFLEX flow cytometer (Brea, CA, USA). Relative fluorescence emissions of gated cells by forward and side scatter properties (FSC/SSC) were analyzed using the CytExpert Software (Beckman Coulter) and expressed as the MFI ratio on the isotype control. Individual values obtained from three independent experiments (*n* = 3) were summarized as means and standard deviations.

### 2.9. ELISA

The concentrations of IL-1β, IL-10, TNF-α, and IL-18 in cell culture supernatants were measured using the commercial ELISA kit (Thermofisher Scientific) according to the instructions of the producer. Plates were scanned using a specialized Charge Coupled Device cooled tool. The integrated density values of the spots of known standards were used to generate a standard curve. Density values for unknown samples were determined using the standard curve for each analysis to calculate the real values in pg/mL. All steps were performed twice and at room temperature. The intra- and inter-assay reproducibility was >90%. Duplicate values that differed from the mean by more than 10% were considered suspect and therefore repeated. We did not report missing values [28].

### 2.10. Statistical Analyses

Each assay was replicated at least three times. Data are expressed as mean ± SD (standard deviation), and statistical significance was determined using the XLStat software, version 2.2 (New York, NY, USA). Statistical analyses for data obtained from monocytes/macrophages were performed by one-way analysis of variance (ANOVA) followed by Tukey’s multiple comparison tests by means of the Prism 5.0 software (GraphPad, San Diego, CA, USA). All results were expressed as mean values ± standard deviations. Values of *p* < 0.05 were considered statistically significant.

## 3. Results

### 3.1. Effect of QMR Treatment on Cell Viability

In order to determine the best exposure time and nominal powers for QMR stimulation, an MTS test was performed at 24 h on both undifferentiated (monocyte) and differentiated (macrophages) THP-1 cell cultures stimulated for 5, 10, and 30 min. The results were compared to non-exposed cells set at 100% cell viability. We found that 10 min of QMF stimulation and 30 nominal power was the optimal culture condition. At higher QMR doses, the cell viability of both cells was lower compared to the controls.

Based on previous results, all subsequent experiments were carried out by exposing the cells to QMR for 10 min, setting up two experimental sets: after 24 h, and repeating 4 times over a period of 4 days (Figure 1). To evaluate the potential cytoprotective effect of QMR exposure, we used an LPS/HA-induced inflammatory cell model. The data presented in Figure 1D,E show that QMR treatment alone did not have a significant effect on cellular viability in the differentiated (macrophages) THP-1 cell being studied. The stimulation with LPS/HA reduced the cell metabolic activity compared to the sham-exposed control cell. QMR treatment was able to restore the metabolic activity to values similar to sham-exposed control cells at both experimental sets (after 24 h (D) and 4 days (E)).

### 3.2. QMR Inhibits Activity and Protein Expression of COX-2 and iNOS

We investigated the effect of QMR treatment on the activity and protein expression of the well-known proinflammatory mediators COX-2 and iNOS after LPS/HA stimulation in differentiated THP-1 cells (Figure 2 and Figure 3). To determine whether QMR can inhibit the COX-2 activity, macrophages THP-1 cells were stimulated with LPS/HA, and the amount of PGE_2_ release was assessed using anti-PGE_2_-coated ELISA plates. Cells stimulated with LPS/HA exhibited a significant increment in PGE_2_ release compared with sham-exposed control cells (CTRL). This induction was significantly reduced (*p* < 0.05) by treatment with QMR at both experimental sets (after 24 h and 4 days) (Figure 2A,B). To confirm the inhibitory effects of QMR treatment on COX-2, LPS/HA-treated macrophage cells were assessed for their protein level, using specific antibodies by Western blotting analysis. As shown in Figure 2, LPS/HA stimulation caused an increased COX-2 protein expression at both times 24 h (A) and 4 days (B), as assessed by measuring the relative COX-2 density compared to sham-exposed cells (*p* < 0.05). The QMR exposure following the LPS/HA stimulation led to a significant reduction in COX-2 protein expression in both experimental sets (*p* < 0.05).

Stimulation of cells with LPS/HA resulted in a significant enhancement of the iNOS activity compared with sham-exposed control cells (Figure 3A,B) at 24 h and 4 days. This indicated increased production of NO and release of its stable product into the culture medium. However, cells stimulated with QMR showed a reduction in the iNOS enzyme activity following stimulation with LPS/HA. Western blot analysis applied to identify iNOS protein showed an iNOS up-regulated after LPS/HA stimulation at both times when examined with respect to sham-exposed cells. In accordance with activity results, QMR treatment produced a significant reduction in LPS/HA-induced iNOS protein expression at all time points studied, with a higher action in the experimental set of 4 days (Figure 3A,B). Moreover, the ability of QMR treatment to inhibit iNOS protein expression was much stronger than the ability to inhibit COX-2. Lastly, as COX-2 and iNOS expression is activated by the transcription factor NF-kB, it was assumed that QMR treatment may also inhibit NF-kB (Figure 4). The protein expression level of NF-kB was studied using p65-specific antibodies. LPS/HA stimulation strongly induced NF-κB activity of the macrophage THP-1 cells by monitoring the levels of p-p65 protein in nuclear extracts by Western blot analysis. The QMR treatment 24 h after cell activation by LPS/HA significantly reduced NF-κB activity compared to sham-exposed control cells (Figure 4A). This reduction remained significant after 4 days of QMR treatment as shown in Figure 4B.

### 3.3. Effect of QMR on Nitrosative Stress

We investigated the effects of QMR treatment on peroxynitrite production and tyrosine nitration in PMA-differentiated THP-1 cells in response to LPS/HA stimulation. Peroxynitrite is a reactive nitrogen species that can form when large amounts of NO and superoxide are present during inflammatory conditions. In this study, intracellular peroxynitrite was detected using fluorescence-based quantitative measurements. Figure 5 showed that peroxynitrite levels were significantly enhanced after the LPS/HA stimulation only 4 days compared to sham-exposed cells. After treatment with QMR, we observed a significant reduction in peroxynitrite levels. Nitration of tyrosine residues as 3-nitrotyrosine is considered a biomarker of peroxynitrite production. In our cellular model, the larger amount of immunoreactive band for 3-nitrotyrosine was present at a molecular weight of about 70–75 kDa. Our results show that after LPS/HA stimulation, nitrated proteins accumulate in a significant manner at 4 days and QMR treatment, in PMA-differentiated THP-1 cells, restored values of tyrosine nitration similar to the sham-exposed control cells at the same time point. No difference was recorded at 24 h (Figure 6). These findings suggest that QMR treatment may have a beneficial effect in reducing peroxynitrite production and nitrosative stress in immune cells.

### 3.4. QMR Treatment Attenuates LPS/HA-Induced Activation of NLRP3 Inflammasome

We next investigated whether QMR treatment affects NLRP3 inflammasome signaling.

Inflammasome signaling is a critical component of the innate immune response to environmental danger signals, which can trigger chronic inflammation in various diseases, including OA. Inflammasomes are multiprotein complexes that detect PAMPs or DAMPs and activate caspase-1, which then cleaves and releases pro-inflammatory cytokines such as IL-1β and IL-18 [29].

To further clarify the effect of QMR treatment on inflammasome activation, we examined the protein expression of the inflammasome signaling pathway. Results showed that the protein expression levels of NLRP3 and activated caspase-1 (Figure 7) were significantly increased after the LPS/HA induction compared to sham-exposed cells. However, the high level of proteins induced by LPS/HA was dramatically decreased in the QMR treatment group in both experimental set points.

The activation of the NLRP3 inflammasome is responsible for the processing and release of pro-inflammatory cytokines IL-18 and IL-1β, which are licensed through caspase-1. In the study, QMR treatment was found to downregulate IL-18 and IL-1β protein expression and secretion induced by LPS/HA at both times in macrophage THP-1 cells (Figure 8). Overall, these findings strongly suggest that QMR inhibits the LPS/HA-triggered activation of the NLRP3 inflammasome in THP-1-derived macrophages.

### 3.5. QMR Treatment Affects the Polarization of Macrophages

Recent pieces of evidence suggest that inflammation of the synovium and polarization of macrophages play a role in the pathogenesis of OA [30]. Traditionally, synovial macrophages can be classified into two main subtypes based on their activation state and function. Classically activated M1 macrophages are primarily involved in the inflammatory response and are activated by pro-inflammatory cytokines, such as interferon-gamma (IFN-γ) and LPS. M1 macrophages produce pro-inflammatory cytokines, such as IL-1β, IL-6, IL-12, and TNF-α, which help to activate and recruit other immune cells to the site of infection or tissue injury. Alternately, activated M2 macrophages are activated by anti-inflammatory cytokines, such as IL-4 and IL-13, and produce high levels of anti-inflammatory cytokines, such as IL-10 and transforming growth factor-beta (TGF-β). M2 macrophages are involved in tissue repair and remodeling, and they promote an anti-inflammatory and immunosuppressive environment [31]. Therefore, we investigated the influence of QMR on the polarization of macrophages (Figure 9A,B). The results of the flow cytometry assay showed that the expression of CD80 in LPS/HA-induced macrophage (M1) was remarkably up-regulated at 4 days compared to control cells. We found that QMR treatment of LPS/HA-stimulated macrophage THP-1 cells for 10 min significantly induced the expression of markers of M2 macrophage phenotype (CD163). This significant switching macrophage M1/M2 polarization effect observed was evidenced by the levels of cytokines released in the culture medium. In particular, the up-regulation of TNF-α induced by LPS/HA stimulation and cytokine related to M1-type was attenuated with a simultaneous increase in the anti-inflammatory cytokine (IL-10) associated with the M2 phenotype due to QMR treatment (Figure 9C,D).

## 4. Discussion

Osteoarthritis is a chronic degenerative joint disease characterized by cartilage breakdown, subchondral bone sclerosis, and synovial inflammation. Inflammation is a key factor in the pathogenesis of OA, and synovial macrophages play a crucial role in mediating this inflammatory response. Inflammation in OA is driven by various factors, including mechanical stress, cartilage degradation products, and cytokines such as IL-1β and TNF-α [32]. Inflammatory cytokines activate synovial macrophages, which are the major producers of pro-inflammatory mediators such as IL-1β, TNF-α, and PGE_2_. Synovial macrophages also play a key role in the activation of the NLRP3 inflammasome, which is a multiprotein complex involved in the maturation and secretion of IL-1β. The activation of the inflammasome in synovial macrophages leads to further amplification of the inflammatory response in OA [33]. The use of biophysical stimulation techniques in clinical medicine has become increasingly popular as a method to promote repair and anabolic activity in tissue or to strengthen the activity of drug treatment while lessening side effects [34]. The complexity of the interaction between physical agents and biological systems made research difficult, but recent developments in physics have led to a greater understanding of how physical means can be used clinically. The new pharmacology involves identifying the effects of physical agents on a particular cell function, which forms the basis of its clinical application. The cell membrane has been identified as a target and site of interaction, and the physical signal activates a cascade of intracellular events through which transduction pathways differ depending on the type of energy used [35].

The promotion of cellular energy production is one of the mechanisms by which QMR therapy could exert anti-inflammatory effects. By enhancing mitochondrial function and increasing ATP production, the therapy can help to reduce cellular stress and inflammation. ATP is a key molecule that provides energy for cellular processes, and its production is closely linked to mitochondrial function. When mitochondrial function is impaired, as can occur in inflammatory conditions, there is a decrease in ATP production, which can lead to further cellular stress and inflammation. QMR therapy may help to counteract this by enhancing mitochondrial function and increasing ATP production, which can in turn reduce cellular stress and inflammation.

While the effect observed may be function-specific rather than cell- or tissue-specific, this allows all conditions positively influenced by the activation or modulation of this cell function to be treated with the same physical agent. Furthermore, PEMFs have shown potential therapeutic effects in the treatment of OA. Although studies in animal models have demonstrated that PEMFs based therapy can improve bone and cartilage turnover, the results of clinical trials in humans have been inconclusive. The conflicting results can be attributed to differences in study design, small sample sizes, and various subject-related factors [36].

This study was conducted to assess the anti-inflammatory activity of QMR technology in vitro. Specifically, we used an in vitro model of osteoarthritis-related inflammation to evaluate the potential mechanisms of QMR intervention and directed at improving therapeutic efficacy. This model involved inducing inflammatory cytokine expression using a combination of LPS and HA fragments such as PAMPs and DAMPs in the THP-1 cell line differentiated in macrophages.

A preliminary MTS test was performed to determine the optimal exposure time and nominal power for QMR stimulation on undifferentiated and differentiated THP-1 cell cultures. It was found that 10 min of QMR stimulation and 30 nominal power was the best culture condition. We observed that QMR treatment alone did not have a significant effect on cellular viability, but it was able to restore the metabolic activity to values similar to the sham-exposed control cells in both experimental sets (after 24 h and 4 days), which suggests a potential cytoprotective effect of QMR exposure.

We reported for the first time that QMR treatment has a beneficial effect in reducing proinflammatory mediators and nitrosative stress in immune cells. The results showed the inhibitory effects of QMR treatment on COX-2 and iNOS protein expression and activity after LPS/HA stimulation. Western blot analysis showed that LPS/HA stimulation caused an increase in COX-2 and iNOS protein expression compared to sham-exposed cells. However, QMR treatment following LPS/HA stimulation led to a significant reduction in COX-2 and iNOS protein expression. Moreover, the study also investigated the effect of QMR treatment on NF-kB, a transcription factor that activates COX-2 and iNOS expression. The results showed that QMR treatment significantly reduced NF-κB activity compared to sham-exposed control cells after both 24 h and 4 days of LPS/HA stimulation. Moreover, QMR treatment led to a significant reduction in peroxynitrite levels, reactive nitrogen species that can form during inflammatory conditions, and restored values of tyrosine nitration similar to sham-exposed control cells.

The main contribution of inflammatory mediators in OA is to promote the breakdown of cartilage. Proinflammatory cytokines such as IL-1β, IL-6, TNF-α, and IL-18 are consistently elevated in OA and participate in chronic and low-grade inflammation, also known as inflammaging. Cartilage breakdown results in the release of cartilage matrix fragments that stimulate chondrocytes, synovial fibroblasts, and immune cells, leading to the release of matrix metalloproteinases and DAMPs, which promote the development of OA [37]. Inflammasomes, multi-protein complexes activated by DAMPs binding to pattern recognition receptors (PPRs), play a critical role in the innate immune response and are a major disruptor of tissue homeostasis in various diseases, including OA. Dysregulation of inflammasome activity promotes the upregulation of proinflammatory cytokines, apoptosis, and inflammation. NLRP3 and NLRP1 are important inflammasome components in OA, triggering pathogenic processes in macrophages. Targeting inflammasomes in chondrocytes and fibroblast-like synoviocytes presents a promising strategy for developing OA disease-modifying therapies [38].

To date, to our best knowledge, there has been no report about the regulation of QMR treatment on IL-1β/NALP3/caspase-1 expression in OA progression. In this study, we compared NALP3/caspase-1 expression in LPS/HA-stimulated human THP-1 monocytes with QMR treatment to the sham group. The findings indicate that the protein expression levels of NLRP3 and activated caspase-1 were significantly increased by LPS/HA induction compared to sham-exposed cells. However, in both experimental set points, the high protein levels induced by LPS/HA were significantly reduced by QMR treatment. The activation of the NLRP3 inflammasome leads to the processing and release of pro-inflammatory cytokines IL-18 and IL-1β, which are licensed via caspase-1. QMR treatment was found to downregulate IL-18 and IL-1β protein expression and secretion induced by LPS/HA at both time points in macrophage THP-1 cells. Overall, these results strongly suggest that QMR inhibits the LPS/HA-triggered activation of the NLRP3 inflammasome in THP-1-derived macrophages.

Recent research has identified inflammasomes as key regulators of macrophage polarization in OA [39]. The infiltration of immune cells, such as macrophages, into the joint space is a hallmark of OA. They can adopt different polarization states, depending on the signals they receive from the local microenvironment. Two main phenotypes have been described: M1 and M2 macrophages [40]. M1 macrophages are classically activated by pro-inflammatory stimuli, such as IFN-γ and LPS, and produce high levels of pro-inflammatory cytokines, such as TNF-α, IL-1β, and IL-6. M1 macrophages also have high levels of iNOS and produce ROS, which can cause tissue damage. In contrast, M2 macrophages are activated by anti-inflammatory cytokines, such as IL-4 and IL-13, and produce high levels of anti-inflammatory cytokines, such as IL-10 and TGF-β. M2 macrophages are also involved in tissue repair and remodeling. In OA, the balance between M1 and M2 macrophages is disrupted, leading to a chronic inflammatory state that contributes to disease progression. Studies have shown that M1 macrophages are more prevalent in the synovial fluid and tissues of OA patients than M2 macrophages. Liu et al. showed that the ratio of M1/M2 macrophages in synovial fluid and peripheral blood was markedly higher in knee OA patients compared to healthy controls and positively correlated with the severity of knee OA based on KL grade [41]. M1 macrophages contribute to joint tissue destruction by producing pro-inflammatory cytokines and degrading extracellular matrix components. They also stimulate the production of metalloproteinases, which can cause cartilage degradation. In contrast, M2 macrophages have anti-inflammatory properties and can promote tissue repair by producing growth factors and extracellular matrix components.

The results of a transwell in vitro study investigated the effects of M1- and M2-polarized macrophages on OA chondrocytes. The study found that coculture of M1-polarized macrophages with OA chondrocytes could exacerbate the severity of OA, whereas coculture of M2-polarized macrophages with OA chondrocytes attenuated expression of OA-related proteases but increased matrix gene expression. These findings suggest that M1 and M2 macrophages have different effects on the cartilage in OA. The factors that regulate macrophage polarization in OA are complex and not yet fully understood [42]. However, several studies have identified potential molecular targets that could be used to modulate macrophage polarization in OA. For example, the activation of peroxisome proliferator-activated receptor gamma has been shown to induce M2 polarization and reduce joint inflammation in OA. Similarly, the inhibition of TLR4 signaling has been shown to reduce M1 polarization and inflammation in OA [43].

Our findings indicate that QMR treatment induces a switch in macrophage polarization from the M1 phenotype to the M2 phenotype. We observed a significant increase in CD80 expression in LPS/HA-induced macrophages, indicating the presence of M1 macrophages. However, when these macrophages were treated with QMR for 10 min, we observed a significant upregulation of CD163 expression, a marker of M2 macrophages, after 4 days. This suggests that QMR treatment can promote the polarization of macrophages towards the anti-inflammatory M2 phenotype. Moreover, we found that QMR treatment was associated with changes in cytokine levels. Specifically, we observed a reduction in TNF-α, a pro-inflammatory cytokine associated with M1 macrophages, and an increase in IL-10, an anti-inflammatory cytokine associated with M2 macrophages. These changes in cytokine levels provide further evidence that QMR treatment can induce a switch in macrophage polarization.

## 5. Conclusions

In summary, our results suggest that QMR treatment in in vitro model of osteoarthritis-related inflammation reduces proinflammatory mediators and nitrosative stress in immune cells by inhibiting COX-2 and iNOS protein expression as well as reducing NF-κB activity and peroxynitrite levels. QMR treatment also downregulates IL-18 and IL-1β protein expression and secretion induced by LPS/HA at both time points in macrophage THP-1 cells. The study provides evidence that QMR technology inhibits the LPS/HA-triggered activation of the NLRP3 inflammasome in THP-1-derived macrophages, suggesting that QMR treatment may have a beneficial effect in reducing inflammation associated with OA.

The study is characterized by common limitations inherent to the use of an in vitro model.

Briefly, the study only examined a limited set of inflammatory markers and pathways affected by QMR technology. Future studies should investigate a broader range of inflammatory markers and pathways to gain a more comprehensive understanding of QMR technology’s anti-inflammatory effects. The study only investigated the effects of QMR technology on macrophages stimulated with lipopolysaccharide and hyaluronic acid fragments. While these stimuli are commonly used to induce inflammation in vitro, they may not fully replicate the complex inflammatory processes that occur in osteoarthritis in vivo. Therefore, future studies should investigate the effects of QMR technology on a broader range of cell types and inflammatory stimuli to determine its potential efficacy in the treatment of osteoarthritis and other inflammatory conditions. Therefore, the study’s findings should be validated in animal models and human clinical trials.

However, the current study is novel and is our first report of the ability of QMR technology in modulating macrophage polarization towards the anti-inflammatory M2 phenotype. It enriches the knowledge that fills the gap in the current in vitro research available in the literature that precedes the in vivo application of QMR for alleviating pain associated with osteoarthritis.

These results are promising and have the potential to provide significant opportunities for the clinical rehabilitative treatment of patients with osteoarthritis and other inflammation-correlated diseases. These findings also have important implications for the development of new therapeutic strategies aimed at targeting macrophage polarization in the treatment of inflammatory diseases.

## Figures and Tables

**Figure 1 antioxidants-12-01358-f001:**
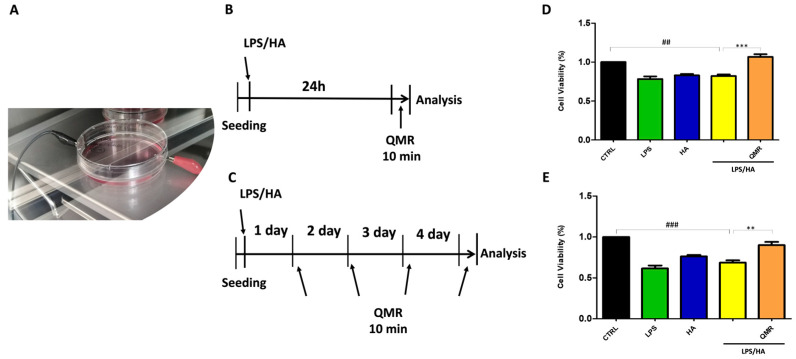
**QMR stimulation protocol.** (**A**) Image of the exposure system. (**B**,**C**) Schematic figure describing the timelines of the experiments. PMA differentiate THP-1 human monocytic cells were stimulated with LPS (10 ng/mL) and HA (10 μg/mL). In panel B, after 24 h, cells were treated with QMR for 10 min. In panel (**C**), cells were treated with QMR 10 min/day for 4 consecutive days at 30 nominal powers. (**D**,**E**) Cellular viability after QMR treatment. Histograms represent the % of cellular viability determined using MTS assay after QMR treatment (10 min) at the two different settings (D: 24 h and E: 4 days). Sham-exposed controls (CTRL) were kept in parallel. Data were represented as mean ± SD of *n* = 6 independent experiments. ^##^
*p* < 0.01 vs. control cells; ^###^
*p* < 0.001 vs. control cells; ** *p* < 0.01 vs. LPS/HA-treated cells; *** *p* < 0.001 vs. LPS/HA-treated cells.

**Figure 2 antioxidants-12-01358-f002:**
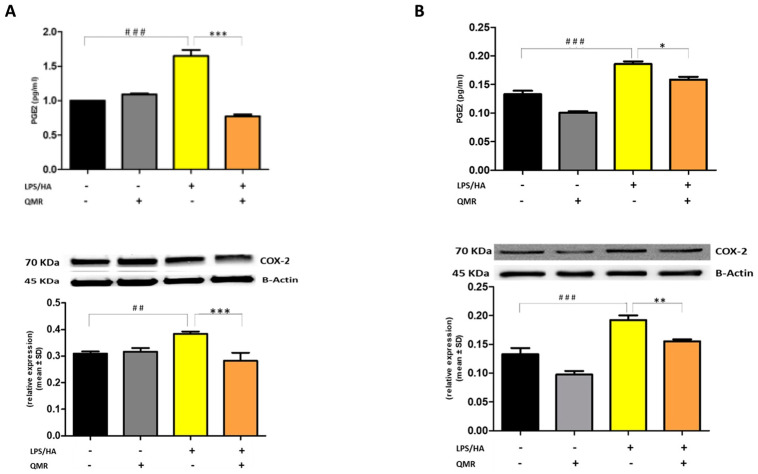
**Effect of QMR treatment on PGE_2_ levels and COX-2 protein expression in differentiated (macrophages) THP-1 cells.** QMR treatment for 10 min at the two different settings (**A**) 24 h and (**B**) 4 days. Prostaglandin (PG) E_2_ levels released in the culture medium are expressed as pg/mL (top). At the bottom, a representative image of Western blot experiments performed on cell lysates and averaged band density of relative expression of COX-2 normalized vs. β-actin. The data represented means ± SD (*n* = 6). ^##^
*p* < 0.01 vs. control cells; ^###^
*p* < 0.001 vs. control cells; * *p* < 0.05 vs. LPS/HA-treated cells; ** *p* < 0.01 vs. LPS/HA-treated cells; *** *p* < 0.001 vs. LPS/HA-treated cells.

**Figure 3 antioxidants-12-01358-f003:**
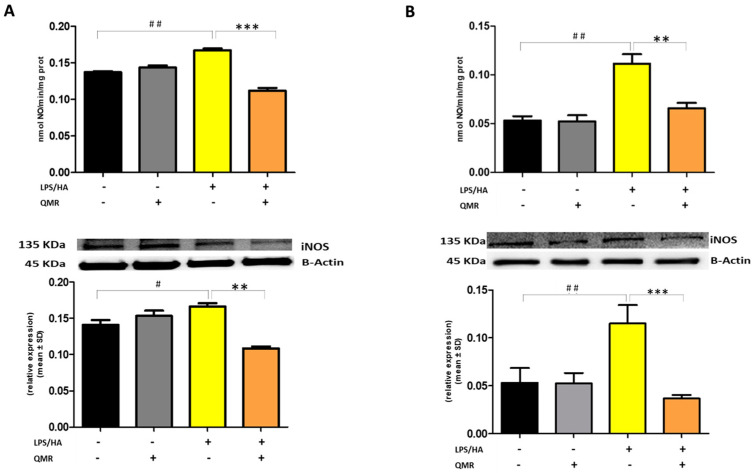
**Effect of QMR treatment on iNOS activity and protein expression in differentiated (macrophages) THP-1 cells.** QMR treatment for 10 min at the two different settings (**A**) 24 h and (**B**) 4 days. Spectrophotometric evaluation of the enzymatic activity of NOS performed on cell lysates and NOS enzymatic activities are expressed in nmol/min/mg prot. (top). At the bottom, a representative image of Western blot experiments performed on cell lysates and averaged band density of relative expression of iNOS normalized vs. β-actin. The data represented means ± SD (*n* = 3). ^#^
*p* < 0.05 vs. control cells; ^##^
*p* < 0.01 vs. control cells; ** *p* < 0.01 vs. LPS/HA-treated cells; *** *p* < 0.001 vs. LPS/HA-treated cells.

**Figure 4 antioxidants-12-01358-f004:**
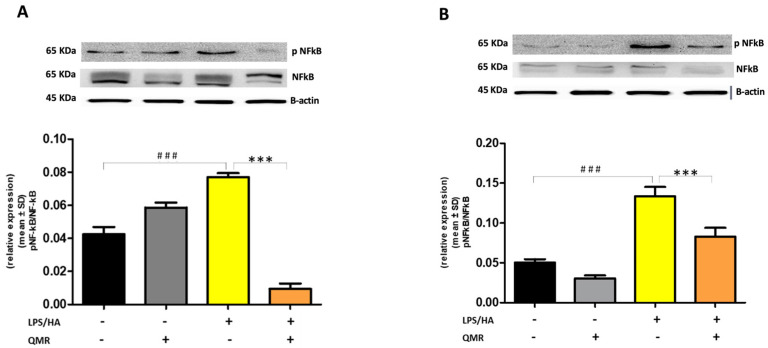
**Effect of QMR treatment on NF-kB expression in differentiated (macrophages) THP-1 cells.** QMR treatment for 10 min at the two different settings (**A**) 24 h and (**B**) 4 days. Representative image of Western blot experiments performed on cell lysates (top). At the bottom, averaged band density of relative expression of pNF-kB (p65) normalized vs. β-actin. The data represented means ± SD (*n* = 3). ^###^
*p* < 0.001 vs. control cells; *** *p* < 0.001 vs. LPS/HA-treated cells.

**Figure 5 antioxidants-12-01358-f005:**
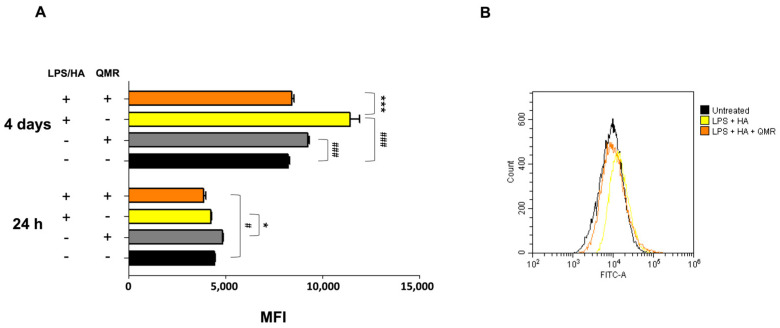
**Generation of peroxynitrites in differentiated (macrophages) THP-1 cells after 24 h and 4 days of QMR treatment.** (**A**) Bar graphs show the fold increase in the mean fluorescence intensities (MFIs) related to the emissions in the FL-1/FITC channel, which is proportional to the generation of peroxynitrite. Values are the ratios of the MFI generated from each sample on the unstained control (negative). (**B**) Peaks of emission were obtained using flow cytometry and are generated by plotting the cell count (y-axis) and the FITC fluorescence emission (x-axis). ^#^
*p* < 0.05 vs. control cells; ^###^
*p* < 0.001 vs. control cells; * *p* < 0.05 vs. LPS/HA-treated cells.; *** *p* < 0.001 vs. LPS/HA-treated cells.

**Figure 6 antioxidants-12-01358-f006:**
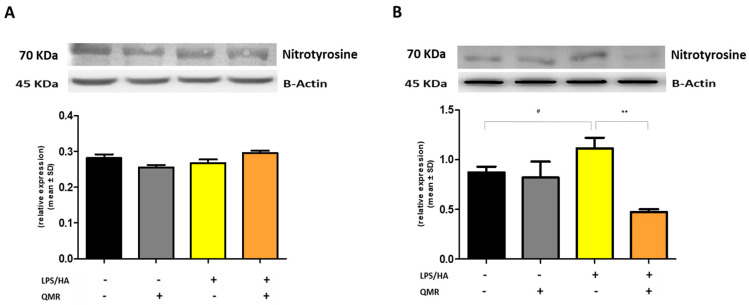
**Nitrotyrosine protein expression in differentiated (macrophages) THP-1 cells.** Effect of QMR treatment (10 min) after 24 h of incubation cells (**A**) and 4 days (**B**) on the expression of the 3-nitrotyrosine protein in LPS/HA-stimulated cells. Representative image of Western blot experiment (top) and at the bottom averaged band density of protein normalized vs. β-actin. The data represented means ± SD (*n* = 3). ^#^
*p* < 0.05 vs. control cells; ** *p* < 0.01 vs. LPS/HA-treated cells.

**Figure 7 antioxidants-12-01358-f007:**
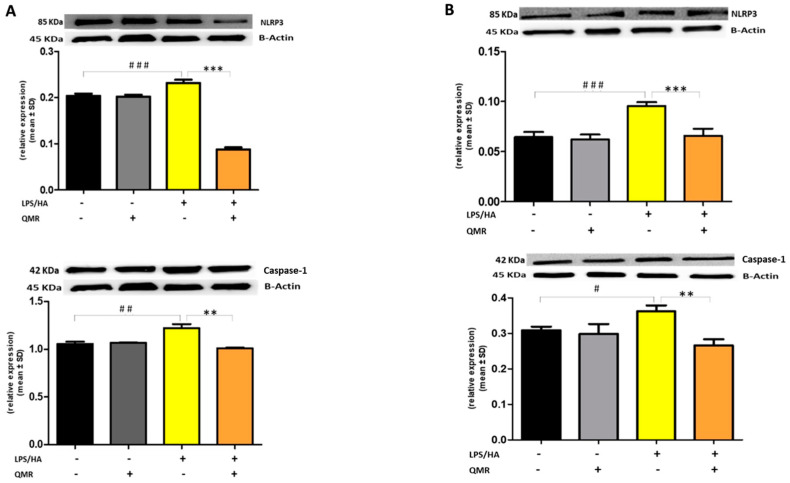
**Effect of QMR treatment on NLRP3 and caspase-1 protein expression in differentiated (macrophages) THP-1 cells.** QMR treatment for 10 min at the two different settings (**A**) 24 h and (**B**) 4 days. Representative image of Western blot experiments performed on cell lysates (top). At the bottom, averaged band density of relative expression of pNF-kB (p65) normalized vs. β-actin. The data represented means ± SD (*n* = 3). ^#^
*p* < 0.05 vs. control cells; ^##^
*p* < 0.01 vs. control cells; ^###^
*p* < 0.001 vs. control cells; ** *p* < 0.01 vs. LPS/HA-treated cells; *** *p* < 0.001 vs. LPS/HA-treated cells.

**Figure 8 antioxidants-12-01358-f008:**
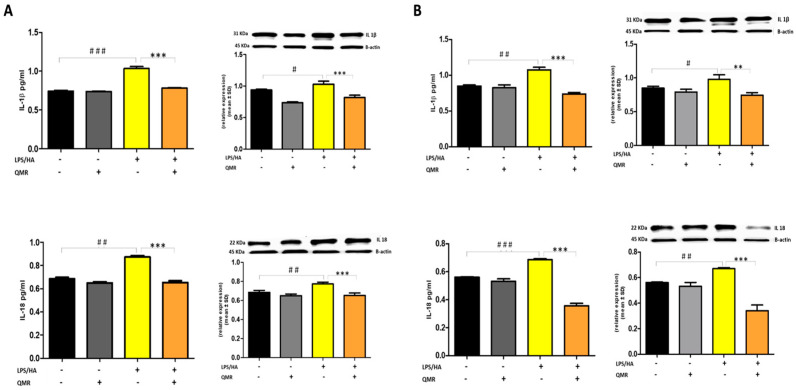
**Effect of QMR treatment on IL-1β and IL-18 protein expression and secretion in differentiated (macrophages) THP-1 cells.** THP-1-derived macrophages were stimulated with LPS/HA and treated with QMR for 10 min at two different settings (**A**) 24 h and (**B**) 4 days. The cytokines IL-1β (top) and IL-18 (bottom) were determined using ELISA kits (left) and relative protein expression normalized vs. β-actin was detected using a Western blot analysis (right). The data represented means ± SD (*n* = 3). ^#^
*p* < 0.05 vs. control cells; ^##^
*p* < 0.01 vs. control cells; ^###^
*p* < 0.001 vs. control cells; ** *p* < 0.01 vs. LPS/HA-treated cells; *** *p* < 0.001 vs. LPS/HA-treated cells.

**Figure 9 antioxidants-12-01358-f009:**
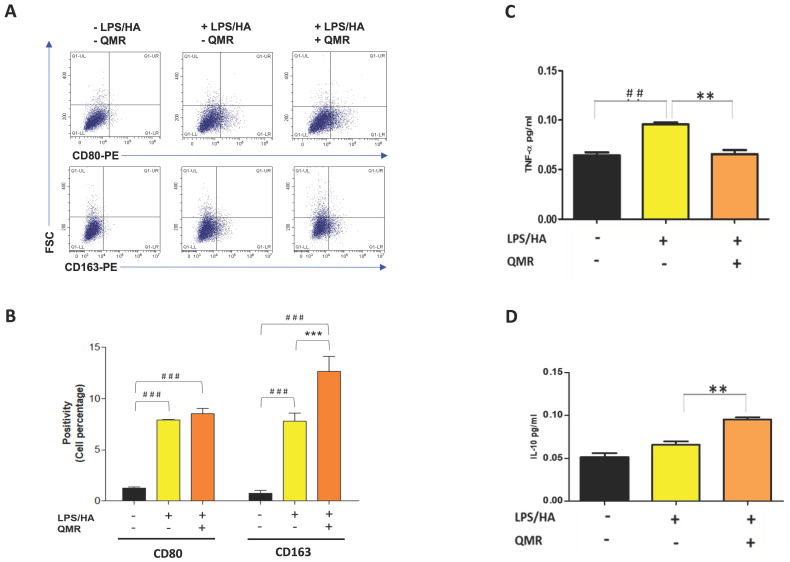
Effect of QMR treatment on polarization-related immunophenotype profile and IL-10 and TNF-α secretions in LPS/HA-stimulated differentiated macrophages. (**A**) Representative dot plots show the distribution of the cell population in response to QMR treatments at 4 days. Dot plots represent the population stained positive for CD80-PE or CD163-PE (x-axis) plotted towards the forward scatter (y-axis: FSC-Width). (**B**) The graph shows the percentages of the cell population stained positive for CD80 or CD163 after 4 days. The cytokines TNF-α (**C**) and IL-10 (**D**) were determined using ELISA kits after 4 days. The data represented means ± SD (*n* = 3). ^##^ *p* < 0.01 vs. control cells; ^###^ *p* < 0.001 vs. control cells; ** *p* < 0.01 vs. LPS/HA-treated cells; *** *p* < 0.001 vs. LPS/HA-treated cells.

## Data Availability

Data are contained within the article.

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
