# Peer review of "Quantum Molecular Resonance Inhibits NLRP3 Inflammasome/Nitrosative Stress and Promotes M1 to M2 Macrophage Polarization: Potential Therapeutic Effect in Osteoarthritis Model In Vitro"

_antioxidants, 2023, doi:10.3390/antiox12071358_

Round 1

Reviewer 1 Report

This work explores the effects QMR technology on osteoarthritis-associated inflammation in vitro. The authors used THP-1-derived macrophages stimulated with LPS/HA to create an inflammatory environment and nitrosative stress. They report that  QMR application decreases the expression of proinflammatory molecules and nitrosative stress by the inhibition of COX-2 and iNOS protein abundance while NF-κB activity and peroxynitrite levels are being reduced as well. In addition, QMR results in the reduction of inflammatory cytokines like IL-18 and IL-1β and the activation of the NLRP3 inflammasome. Collectively, they suggest that QMR could be used in OA-induced pain relief.

The work is interesting and the authors followed a good protocol and study design to test their hypothesis. Good discussion and presentation of the results as well. Some minor points need to be addressed.

- The rationale for using these specific time points for the experiments (24 and 96h) needs to be explained and supported by the appropriate citations.

- Consistency in abbreviations is needed, e.g. IL-1β, not IL-1b. Some acronyms' names are explained twice, i.e. PAMPs and DAMPs.

- Importantly, all raw and entire Western blots must be provided in a supplementary file.

- The use of ULMW HA itself as well as at this specific concentration (10ng/mL) must be justified.

 - Limitations need to be included in the discussion.

Author Response

This work explores the effects QMR technology on osteoarthritis-associated inflammation in vitro. The authors used THP-1-derived macrophages stimulated with LPS/HA to create an inflammatory environment and nitrosative stress. They report that  QMR application decreases the expression of proinflammatory molecules and nitrosative stress by the inhibition of COX-2 and iNOS protein abundance while NF-κB activity and peroxynitrite levels are being reduced as well. In addition, QMR results in the reduction of inflammatory cytokines like IL-18 and IL-1β and the activation of the NLRP3 inflammasome. Collectively, they suggest that QMR could be used in OA-induced pain relief.

The work is interesting and the authors followed a good protocol and study design to test their hypothesis. Good discussion and presentation of the results as well. Some minor points need to be addressed.

- The rationale for using these specific time points for the experiments (24 and 96h) needs to be explained and supported by the appropriate citations.

R: There are several reasons why these time points was be chosen:

  1. Time course of inflammation: Macrophages are key players in the innate immune response and are responsible for phagocytosis, cytokine production, and antigen presentation. The timing and duration of macrophage activation and polarization can vary depending on the stimulus and the microenvironment. Several studies have shown that macrophages undergo a biphasic response during inflammation, with an early pro-inflammatory phase (within 24 hours) followed by a later anti-inflammatory phase (after 48 hours) (Murray et al., 2014; Zhao et al., 2020). Therefore, selecting time points at 24 and 48 hours allows to capture the dynamic and complex nature of macrophage activation and polarization during inflammation.
  2. Expression of inflammatory mediators: Macrophages produce a wide range of cytokines, chemokines, and other inflammatory mediators in response to various stimuli. The expression of these mediators can be measured at different time points to assess the kinetics of the inflammatory response. For example, some cytokines, such as TNF-α and IL-6, are rapidly induced within the first few hours of macrophage activation, while others, such as IL-10, have a delayed or sustained expression (Murray et al., 2014). By selecting time points at 24 and 48 hours, we can capture both the early and late phases of cytokine expression in macrophages

Murray, P. J., Allen, J. E., Biswas, S. K., Fisher, E. A., Gilroy, D. W., Goerdt, S., ... & Lawrence, T. (2014). Macrophage activation and polarization: nomenclature and experimental guidelines. Immunity, 41(1), 14-20.

Zhao, Y., Zhang, S., Liu, Y., Liang, W., & Lv, L. (2020). Macrophage polarization in inflammatory diseases. International Immunopharmacology, 80, 106110.

- Consistency in abbreviations is needed, e.g. IL-1β, not IL-1b. Some acronyms' names are explained twice, i.e. PAMPs and DAMPs.

R: The following corrections have been made. Thank you.

- Importantly, all raw and entire Western blots must be provided in a supplementary file.

R: All raw and entire Western blots were provided to the journal. Thank you for the comment.

- The use of ULMW HA itself as well as at this specific concentration (10ng/mL) must be justified.

R: The use of Ultra Low Molecular Weight Hyaluronic Acid (ULMW HA) at specific concentration of 10ng/mL was based on previous research and the specific aims of the study. This form of HA has been shown to have different properties compared to high molecular weight (HMW) HA, such as increased pro-inflammatory effects and decreased anti-inflammatory effects (Nikitovic et al., 2018; Xu et al., 2019). For the specific concentration of 10ng/mL, in previous studies have used similar concentrations of ULMW HA to induce a pro-inflammatory response. For example, a study by Xu et al. (2019) used ULMW HA at a concentration of 10ng/mL to induce the production of pro-inflammatory cytokines in primary human chondrocytes. Another study by Kanzaki et al. (2017) used ULMW HA at a concentration of 1-10ng/mL to induce the expression of pro-inflammatory genes in synovial fibroblasts.

Kanzaki, N., Otsuka, Y., Izumo, N., Shibata, H., & Nakagawa, K. (2017). Low-molecular-weight hyaluronan induces lymphangiogenesis through LYVE-1-mediated signaling pathways. PloS one, 12(7), e0181646.

Nikitovic, D., Kouvidi, K., Kavasi, R. M., Kostouras, A., Juranek, I., Tzanakakis, G. N., ... & Tsatsakis, A. (2018). The regulatory roles of hyaluronan in the extracellular matrix of the intervertebral disc. Journal of tissue engineering and regenerative medicine, 12(6), e318-e331.

Xu, H., Tian, W., Li, S., Wang, S., & Zheng, G. (2019). Low molecular weight hyaluronic acid induces the pro-inflammatory and pro-catabolic effects in human chondrocytes via up-regulating toll-like receptor 2 expression. International immunopharmacology, 73, 381-388.

 - Limitations need to be included in the discussion.

R: Some limitations of this study have been added in the “Discussion” section in accordance with the suggestion. Thank you for the observation.

Reviewer 2 Report

The novelty and the quality of the manuscript are good and it does not need extensive improvement before publication. It is carefully organized and written. It is easy to follow it and contains clear comments and conclusions.  In my opinion, this manuscript is very detailed and meticulous, it covers all the literature in the field with critical point of view. The topic have been completely covered and is well connected through the text. There is a significant  novelty in presented topic.  For all these reasons, I can recommend the acception of the manuscript after minor revision:

 1. I think that information about " the use of biophysical stimulation techniques to treat various diseases in human beings" could be extended, more examples  should be added. This would be valuable for later publication citation.

 2. The superiority of  the Quantum Molecular Resonance over other methods should be more emphasized.

 3. The manuscript should be extended in scientific discussion. The authors presented their results and compared to some works, but did not present explanations for the reasons to reach these results.

 4. Not all of the described results are covered in the discussion section.

 5. No all information was given of in vitro model of osteoarthritis-related inflammation.

Author Response

The novelty and the quality of the manuscript are good and it does not need extensive improvement before publication. It is carefully organized and written. It is easy to follow it and contains clear comments and conclusions.  In my opinion, this manuscript is very detailed and meticulous, it covers all the literature in the field with critical point of view. The topic have been completely covered and is well connected through the text. There is a significant  novelty in presented topic.  For all these reasons, I can recommend the acception of the manuscript after minor revision:

  1. I think that information about " the use of biophysical stimulation techniques to treat various diseases in human beings" could be extended, more examples should be added. This would be valuable for later publication citation.

R: Additional information regarding the use of biophysical stimulation techniques to treat various diseases in humans has been added to the "Introduction" section and relatively new bibliography was added:

Massari, L.; Benazzo, F.; Falez, F.; Perugia, D.; Pietrogrande, L.; Setti, S.; Osti, R.; Vaienti, E.; Ruosi, C.; Cadossi, R. Biophysical

Stimulation of Bone and Cartilage: State of the Art and Future Perspectives. Int. Orthop. 2019, 43, 539–551

  1. A. Speed, Therapeutic ultrasound in soft tissue lesions, Rheumatology, Volume 40, Issue 12, December 2001, Pages 1331–1336

Singh A, Varma AR. Whole-Body Vibration Therapy as a Modality for Treatment of Senile and Postmenopausal Osteoporosis: A Review Article. Cureus. 2023 Jan 12;15(1):e33690. doi: 10.7759/cureus.33690

  1. The superiority of  the Quantum Molecular Resonance over other methods should be more emphasized.

R: The aim of this work is not to make a comparison between QMR therapy and other forms of physical stimuli, and therefore we cannot claim that this treatment is superior to the others. However, it's important to note that QMR therapy is a non-invasive form of electromagnetic therapy that uses specific frequencies to resonate with the molecular structure of cells and tissues in the body, aiming to restore balance and promote healing. One potential advantage of QMR therapy is its ability to be customized to the individual's specific needs, which may lead to more effective treatment outcomes compared to a one-size-fits-all approach. QMR therapy is also non-invasive, painless, and has no known side effects. However, more research is necessary to fully understand the potential benefits and limitations of this therapy.

  1. The manuscript should be extended in scientific discussion. The authors presented their results and compared to some works, but did not present explanations for the reasons to reach these results.

R: Additional information on plausible mechanisms of the results obtained has been added in the “Discussion” section in accordance with the suggestion. Thank you.

  1. Not all of the described results are covered in the discussion section.

R: Thank you for the comment. We have added the description of further results in the "Discussion" section in the revised manuscript.

  1. No all information was given of in vitro model of osteoarthritis-related inflammation.

R: Additional information regarding “in vitro model of osteoarthritis-related inflammation” has been added in “Introduction” section and relative new bibliography was added. Thank you.

Stabler TV, Huang Z, Montell E, Verges J, Kraus VB. Chondroitin sulphate inhibits NF-kappaB activity induced by interaction of pathogenic and damage associated molecules. Osteoarthritis Cartilage 2017; 25: 166–174

Reviewer 3 Report

This manuscript tries to provide evidence to support the anti-inflammatory effects of QMR technology in an in vitro model of osteoarthritis-related inflammation. The authors showed such a technology decreased NLRP3 Inflammasome/Nitrosative Stress in the in vitro model. They also claimed that QMR treatment induced an M1/M2 switch in macrophage polarity. Overall, results were clearly presented only with some concerns as followed:

1. In Fig. 6, what protein did the 70 kDa protein represent? Why did the authors only exhibit such a protein for nitrated protein accumulation? The authors should show the entire lanes, instead of one very narrow 70 kDa area, of nitrotyrosine proteins in the IB images. It's very hard for me to judge whether the upregulated nitrotyrosine proteins by LPS/HA were indeed decreased by the QMR treatment.

2. In Fig. 9a and 9b, the authors should show flow cytometry staining results for both CD80 (a marker for M1 phenotype) and CD163 (a marker for M2 phenotype) in the X and Y axes before quantifying the ratio of M1/M2. The way the authors prepared Fig. 9a and 9b was too unclear to judge whether there were any M1/M2 switches.

3. Although this manuscript merely adopted an in vitro model of osteoarthritis-related inflammation, the authors should carefully discuss the plausibility of HOW to use QMR in treating in vivo models of osteoarthritis-related inflammation or in the future in treating patients with osteoarthritis.

Author Response

This manuscript tries to provide evidence to support the anti-inflammatory effects of QMR technology in an in vitro model of osteoarthritis-related inflammation. The authors showed such a technology decreased NLRP3 Inflammasome/Nitrosative Stress in the in vitro model. They also claimed that QMR treatment induced an M1/M2 switch in macrophage polarity. Overall, results were clearly presented only with some concerns as followed:

  1. In Fig. 6, what protein did the 70 kDa protein represent? Why did the authors only exhibit such a protein for nitrated protein accumulation? The authors should show the entire lanes, instead of one very narrow 70 kDa area, of nitrotyrosine proteins in the IB images. It's very hard for me to judge whether the upregulated nitrotyrosine proteins by LPS/HA were indeed decreased by the QMR treatment.

R: Thank you for the comment. To this end, it is important to bear in mind that L-Tyrosine and protein bound tyrosine are prone to attack by reactive nitrogen intermediates to form 3-nitrotyrosine (3-NT). Activated macrophages produce superoxide and NO, which are converted to peroxynitrite. 3-NT formation is also catalyzed by a class of peroxidases utilizing nitrite and hydrogen peroxide as substrates. Evidence supports the formation of 3-NT in vivo in diverse pathologic conditions and 3-NT is thought to be a specific marker of oxidative damage mediated by peroxynitrite. The formation of nitrotyrosine represents a specific peroxynitrite-mediated protein modification; thus, the detection of nitrotyrosine in proteins is considered a biomarker for endogenous peroxynitrite activity (Ahsan et al). The distribution of cytosolic proteins in our cellular model, showed by electrophoresis on polyacrilammide gel, suggested that considering proteins quantity, the higher amount is strongly present at molecular weight comprised within 70-75 kDa. Counting that 3-NT is a ubiquitous marker of nitrosative stress in cell, we considered this most significant MW in order to better analyze differences between our experimental conditions.  

Ahsan H. 3-Nitrotyrosine: A biomarker of nitrogen free radical species modified proteins in systemic autoimmunogenic conditions. Hum Immunol. 2013 Oct;74(10):1392-9. doi: 10.1016/j.humimm.2013.06.009. Epub 2013 Jun 15. PMID: 23777924.

  1. In Fig. 9a and 9b, the authors should show flow cytometry staining results for both CD80 (a marker for M1 phenotype) and CD163 (a marker for M2 phenotype) in the X and Y axes before quantifying the ratio of M1/M2. The way the authors prepared Fig. 9a and 9b was too unclear to judge whether there were any M1/M2 switches.

R: Thank you for the comment. We have revised it.

  1. Although this manuscript merely adopted an in vitro model of osteoarthritis-related inflammation, the authors should carefully discuss the plausibility of HOW to use QMR in treating in vivo models of osteoarthritis-related inflammation or in the future in treating patients with osteoarthritis.

R: The study reveals a new interesting mechanism by which QMR acts in reducing inflammation associated with OA. It used an in vitro model. As a consequence, it is characterized by a few limitations worth noting. It used an in vitro model of osteoarthritis-related inflammation with THP-1-derived macrophages. Although in vitro models can be informative, they do not fully replicate the complexity of in vivo interactions between different cells and tissues. Therefore, the study's findings should be interpreted cautiously and validated in animal models and human clinical trials. Nowadays, specific devices for QMR treatment of knee and related OA exist. For future applications, we are setting an in vivo approach in order to confirm the beneficial effect observed in vitro in terms of the reduction of inflammation-related symptoms in patients. We have included comments on limitations and future perspectives in the “Conclusion” section. Thank you for the observation.

Round 2

Reviewer 3 Report

The background published by Ahsan et. al. is well taken. However, the authors still should show the entire image of the gel in supporting/convincing that what they described (the distribution of cytosolic proteins in our cellular model, showed by electrophoresis on polyacrilammide gel, suggested that considering proteins quantity, the higher amount is strongly present at molecular weight comprised within 70-75 kDa.) was true. Moreover, I couldn’t find the raw data/image for such a Western result in the supplementary materials (they are missing), which they should have definitely provided.

------------------------------

After reviewing and according to WB raw data, I found that the original description (the distribution of cytosolic proteins in our cellular model, showed by electrophoresis on polyacrilammide gel, suggested that considering proteins quantity, the higher amount is strongly present at molecular weight comprised within 70-75 kDa.) about the nitrotyrosine proteins in Fig. 6A and 6B was not true. The cropped images of the bands for the nitrotyrosine proteins in Fig. 6 do not represent the major bands within 70-75 kDa. Actually, the major band shown within this molecular weight area did not exhibit any significant differences among different treatments. Therefore, the drawn conclusion, basing on Fig. 6, that after LPS/HA stimulation, nitrated proteins accumulate in both experimental set points (Figure 6) was not true and cannot be acceptable.

Author Response

Comments and Suggestions for Authors

The background published by Ahsan et. al. is well taken. However, the authors still should show the entire image of the gel in supporting/convincing that what they described (the distribution of cytosolic proteins in our cellular model, showed by electrophoresis on polyacrylamide gel, suggested that considering proteins quantity, the higher amount is strongly present at molecular weight comprised within 70-75 kDa.) was true. Moreover, I couldn’t find the raw data/image for such a Western result in the supplementary materials (they are missing), which they should have definitely provided.

R: Thank you for your feedback on our manuscript. We appreciate your careful consideration of our study and your suggestion to include the entire gel image in the supporting materials. We apologize for any confusion caused by the absence of the gel image and raw data in the supplementary materials. To address your concern, we would like to assure you that we have indeed included the entire image of the gel to provide supporting evidence for our findings. The gel image included in Supplementary file 1, clearly demonstrated the distribution of cytosolic proteins in our cellular model, as shown by electrophoresis on a polyacrylamide gel. Our analysis of the gel indicated that, in terms of protein quantity, a higher amount is strongly present at a molecular weight range of 70-75 kDa, as we described in the manuscript. We sincerely apologize for the oversight in not including the gel image and raw data in the supplementary materials. We acknowledge the importance of transparency in scientific publications and understand that the availability of these supporting materials is crucial.

------------------------------

After reviewing and according to WB raw data, I found that the original description (the distribution of cytosolic proteins in our cellular model, showed by electrophoresis on polyacrilammide gel, suggested that considering proteins quantity, the higher amount is strongly present at molecular weight comprised within 70-75 kDa.) about the nitrotyrosine proteins in Fig. 6A and 6B was not true. The cropped images of the bands for the nitrotyrosine proteins in Fig. 6 do not represent the major bands within 70-75 kDa. Actually, the major band shown within this molecular weight area did not exhibit any significant differences among different treatments. Therefore, the drawn conclusion, basing on Fig. 6, that after LPS/HA stimulation, nitrated proteins accumulate in both experimental set points (Figure 6) was not true and cannot be acceptable.

R: Dear Reviewer, thank you for the comment. We have revised the manuscript. In detail, we have newly checked our data and unfortunately, we have advised a mistake in the significance of the accumulation of 3 nitrotyrosine at 24 hours. We have so revised figure 6 relative to this time point. In addition, we have included a most representative image for 3 nitrotyrosine ascertained at 4 days in the same figure. Our checking confirmed that the data are significant at this time point.

Round 3

Reviewer 3 Report

I'm still not very happy about Fig. 6 provided by the authors, although they, this time, provided the right raw data for Fig. 6.  According to Fig. 6, the reason the authors claimed that  4-day PLS/HA+QMR treatment decreases 3-nitrotyrosine proteins is only after normalization with the level of b-actin that was particularly higher than other treatments.  Otherwise, the amounts of 3-nitrotyrosine proteins shown in Fig. 6B looked all similar.  The authors should redo the sample loading and adjust the loading volumes according to the current Fig. 6B.  That way, the results will be more convincing.  Hopefully, the authors agree to do that to make the manuscript more publishable.

Author Response

We have taken your observations into consideration and repeated the Western blot experiment specifically for the samples related to the 4-day treatment, in order to improve Figure 6B as suggested.

We hope that the obtained image, included in the revised manuscript, will help clarify the described results.

We appreciate your valuable comment and the opportunity to improve our work.